# A Phase 1, Randomized, Double-Blinded, Placebo-Controlled and Dose-Escalation Study to Evaluate the Safety and Immunogenicity of the Intranasal DelNS1-nCoV-RBD LAIV for COVID-19 in Healthy Adults

**DOI:** 10.3390/vaccines11040723

**Published:** 2023-03-24

**Authors:** Ruiqi Zhang, Kwok-Hung Chan, Pui Wang, Runhong Zhou, Henry Kwong-Chi Yau, Creany Ka-Wai Wong, Meena Wai-Lam Au, Anthony Raymond Tam, Chi-Tao Ng, Matthew Kwok-Chung Lou, Na Liu, Haode Huang, Shaofeng Deng, Rachel Chun-Yee Tam, Ying Liu, Teng Long, Hoi-Wah Tsoi, Miko K. W. Ng, Jian-Piao Cai, Kelvin Kai-Wang To, Man-Fung Yuen, Zhiwei Chen, Honglin Chen, Kwok-Yung Yuen, Ivan Fan-Ngai Hung

**Affiliations:** 1Department of Medicine, Li Ka Shing Faculty of Medicine, The University of Hong Kong, Pokfulam, Hong Kong SAR, China; 2State Key Laboratory for Emerging Infectious Diseases, Department of Microbiology, Li Ka Shing Faculty of Medicine, The University of Hong Kong, Pokfulam, Hong Kong SAR, China; 3AIDS Institute, Li Ka Shing Faculty of Medicine, The University of Hong Kong, Pokfulam, Hong Kong SAR, China; 4Clinical Trials Centre, Queen Mary Hospital, The University of Hong Kong, Pokfulam, Hong Kong SAR, China

**Keywords:** phase-1, intranasal, DelNS1-nCoV-RBD LAIV, COVID-19

## Abstract

An intranasal COVID-19 vaccine, DelNS1-based RBD vaccines composed of H1N1 subtype (DelNS1-nCoV-RBD LAIV) was developed to evaluate the safety and immunogenicity in healthy adults. We conducted a phase 1 randomized, double-blinded, placebo-controlled study on healthy participants, age 18–55 and COVID-19 vaccines naïve, between March and September 2021. Participants were enrolled and randomly assigned (2:2:1) into the low and high dose DelNS1-nCoV-RBD LAIV manufactured in chicken embryonated eggs or placebo groups. The low and high-dose vaccine were composed of 1 × 10^7^ EID_50_/ dose and 1 × 10^7.7^ EID_50_/ dose in 0.2 mL respectively. The placebo vaccine was composed of inert excipients/dose in 0.2 mL. Recruited participants were administered the vaccine intranasally on day 0 and day 28. The primary end-point was the safety of the vaccine. The secondary endpoints included cellular, humoral, and mucosal immune responses post-vaccination at pre-specified time-points. The cellular response was measured by the T-cell ELISpot assay. The humoral response was measured by the serum anti-RBD IgG and live-virus neutralizing antibody against SARS-CoV-2. The saliva total Ig antibody responses in mucosal secretion against SARS-CoV-2 RBD was also assessed. Twenty-nine healthy Chinese participants were vaccinated (low-dose: 11; high-dose: 12 and placebo: 6). The median age was 26 years. Twenty participants (69%) were male. No participant was discontinued due to an adverse event or COVID-19 infection during the clinical trial. There was no significant difference in the incidence of adverse events (*p* = 0.620). For the T-cell response elicited after full vaccination, the positive PBMC in the high-dose group increased to 12.5 SFU/10^6^ PMBC (day 42) from 0 (baseline), while it increased to 5 SFU/10^6^ PBMC (day 42) from 2.5 SFU/10^6^ PBMC (baseline) in the placebo group. The high-dose group showed a slightly higher level of mucosal Ig than the control group after receiving two doses of the vaccine (day 31, 0.24 vs. 0.21, *p* = 0.046; day 56 0.31 vs. 0.15, *p* = 0.45). There was no difference in the T-cell and saliva Ig response between the low-dose and placebo groups. The serum anti-RBD IgG and live virus neutralizing antibody against SARS-CoV-2 were undetectable in all samples. The high-dose intranasal DelNS1-nCoV-RBD LAIV is safe with moderate mucosal immunogenicity. A phase-2 booster trial with a two-dose regimen of the high-dose intranasal DelNS1-nCoV-RBD LAIV is warranted.

## 1. Introduction

This global pandemic of coronavirus disease 2019 (COVID-19) is caused by the severe acute respiratory syndrome coronavirus 2 (SARS-CoV-2). December 2021 marked the second anniversary of this ongoing pandemic, which has already infected more than 510 million people with 6.2 million deaths, making it one of the deadliest in history [1]. With the lack of specific antiviral treatment, vaccines remain the most promising preventive measures to contain this pandemic.

As of May 2022, 9.7 billion COVID-19 vaccine doses have been administered [1], with more than 110 different vaccines in clinical trials and 10 vaccines already approved by the World Health Organization (WHO) for emergency use [2]. Despite demonstrating satisfactory clinical efficacy and protection against severe diseases and mortality, the current vaccines failed to provide protection against infection and transmission [3,4,5]. New vaccine platforms are needed.

The continuing emergence of variants due to accumulation of mutations in the spike protein have resulted not only in a higher transmission rate but also impacted on the effectiveness of the COVID-19 vaccines [6,7,8]. Besides, no study on the aforementioned vaccines has demonstrated an effective prevention against the transmission of SARS-CoV-2.

A recent phase 1 study on a nebulized viral vector vaccine has demonstrated a satisfactory immunogenic response among the vaccines [9]. Theoretically, intranasal vaccines have the advantage of stimulating mucosal immunoglobulin (Ig) to prevent both viral transmission and infection. Nevertheless, the nebulized viral vector vaccine carried the risk of aerosol generation and transmission of both the vaccine vector and antigens. Intranasal vaccine will be able to offer a safer route of delivery. We reported previously that intranasal vaccination with DelNS1-LAIV provided complete protection against both homologous virus A(H1N1) pdm09 and heterologous H7N9/H5N1 avian influenza challenges in mice [10]. DelNS1-LAIV, one type of live attenuated influenza virus vaccine (LAIV), is rendered non-pathogenic by removing the virulence factor NS1 protein from 2009 H1N1 (A/CA/04/2009). The DelNS1-LAIV replicated better at 33 °C than at 37 °C, and the replication was slightly slower than wild type (A/CA/04/2009) in chicken embryo, which is used to produce influenza vaccine, at 33 °C. Compared with traditional LAIV, DelNS1-LAIV can work as a vector to express antigens from influenza virus or other viruses [10]. Adopting the same technique, a DelNS1-based receptor binding domain (RBD) vaccine– namely DelNS1-nCoV-RBD LAIV—was made [11]. The COVID-19 vaccine elicits an immune response against SARS-CoV-2 by the expression of RBD which is an important antigen of SARS-CoV-2 and target of neutralizing antibody. The vaccine is also delivered intranasally. The purpose of this study is to evaluate the safety and immunogenicity of DelNS1-nCoV-RBD LAIV for COVID-19 in healthy adults.

## 2. Materials and Methods

### 2.1. Study Design

We conducted a phase 1, randomised, double-blinded, placebo-controlled, dose-escalation study to evaluate the safety and immunogenicity of DelNS1-nCoV-RBD LAIV (Figure 1) in healthy adults at the Phase 1 Clinical Trials Centre of The University of Hong Kong located at Queen Mary Hospital, Hong Kong. The study design investigated a dose-escalation approach to evaluate the safety and immunogenicity of DelNS1-nCoV-RBD LAIV for COVID-19 in healthy adults at two dose levels. The study was conducted in compliance with the Declaration of Helsinki and ICH Guideline for Good Clinical Practice E6(R2). The trial protocol was reviewed and approved by the Hong Kong Department of Health and the Institutional Review Board of The University of Hong Kong/Hospital Authority Hong Kong West Cluster (UW 21-054) and was registered at clinicaltrial.gov (NCT04809389).

### 2.2. Participants

Each participant received two vaccinations 4 weeks apart at one of the two dose levels [i.e., 1 × 10^7^ Egg infective dose at 50% (EID_50_) and 1 × 10^7.7^ EID_50_ per 0.2 mL vaccination] or matching placebo. Vaccination started from the low dose level. The first five participants in the low dose cohort entered the sentinel group, whilst the other nine participants at the same dose level were dosed after all sentinel participants completed the Day 8 visit and without meeting the study specific suspension criteria. An independent safety review committee reviewed all available safety data collected from the 14 participants and decided on escalation to the next dose level.

Participants between 18 and 55 years of age underwent a screening visit where a full medical history and physical examination were taken in addition to blood and urine tests. All participants were tested 4 days prior to the first and second vaccination and at each follow-up visit by deep throat saliva RT-PCR. Participants with positive SARS-CoV-2 RT-PCR would be excluded. Key exclusion criteria were known infection with SARS-CoV-2; any significant respiratory or cardiovascular disease; an immunocompromised condition or a history of autoimmune disease; history of severe allergic reaction to any vaccine or substance or hypersensitivity to eggs, egg proteins or gentamicin sulfate; any nasal abnormality that might affect vaccine administration; and positive result on serum antibody test for SARS-CoV-2 within 4 days prior to the first vaccination. Detailed inclusion and exclusion criteria could be found in the Appendix A.

### 2.3. Study Procedures

Each dose level of DelNS1-nCoV-RBD LAIV was provided as a sterile liquid suspension in single-use intranasal sprayer of 0.2 mL. The placebo contained the same excipients used for the DelNS1-nCoV-RBD LAIV but without the active substance. Eligible participants were randomised in a 4:1 ratio to receive either low-dose DelNS1-nCoV-RBD LAIV or placebo, using block randomization at each dose level. The study statistical programmer generated the randomization list and uploaded to a web-based randomization application. Designated unblinded pharmacists dispensed the investigational vaccine (DelNS1-nCoV-RBD LAIV) or placebo according to the treatment group assigned via the randomization application, and research nurses administered 0.1 mL of DelNS1-nCoV-RBD LAIV or placebo into each nostril of each participant, that is a total of 0.2 mL/vaccination. Participants, investigators, research nurses, study coordinators, and other study-related personnel were blinded to treatment group allocation and the unblinded study personnel were prohibited from disclosing allocation information to them.

### 2.4. Study Objectives and End-Points

#### 2.4.1. Safety

The primary endpoints were the incidence of adverse events (solicited local and systemic events) for a 14-day period after the first or second vaccination and unsolicited adverse events within 28 days after receiving each vaccination. Any adverse events of special interest and serious adverse events occurred after vaccination were also recorded.

Participants were required to stay on-site for safety observation for 2 h after each vaccination. Each of them was given a subject diary and an oral thermometer to record any solicited adverse event that occurred for 14 days after each vaccination. Solicited local adverse events included nasal irritation, sneezing, nasal congestion, cough, sore throat, change in smell, change in taste, change in vision, and eye pain. Solicited systemic adverse events included fever, headache, malaise, myalgia, joint pain, nausea, vomiting, diarrhoea, abdominal pain, chills, and sweating.

Adverse events were graded by taking reference to the FDA’s guidance—“Toxicity Grading Scale for Healthy Adult and Adolescent Volunteers Enrolled in Preventive Vaccine Clinical Trials” (Sep 2007) or else by study-specific definitions. Unsolicited adverse events were graded according to Common Terminology Criteria for Adverse Events (CTCAE), version 5.0 [12].

#### 2.4.2. Immunogenicity

The pre-specified secondary endpoints included cellular and humoral responses post-vaccination and the exploratory secondary endpoints included the mucosal saliva Ig immune response. Serum samples were collected for evaluating cellular and humoral immune responses on prespecified days at baseline (day 0) and at 7, 14, 28, 35, and 42 days after the first vaccination. IgG antibody responses against the receptor-binding domain of spike protein (RBD-IgG) of SARS-CoV-2 were assessed by chemiluminescent microparticle immunoassay (CMIA). The neutralizing antibody titres were assessed by live virus microneutralisation (MN) assay in serum samples. T-cell responses against SARS-CoV-2 RBD peptide pool (15-mer overlapping by 11, spanning the whole RBD sequence) were measured by IFN-γ ELISpot assay [13,14]. The PMA/ionomycin was used as the positive control, while anti-CD28/anti-CD49d mAbs treatment as the negative control. Saliva samples were collected for evaluating mucosal immune responses on prespecified days at baseline (day 0) and at 3, 28, 31, and 56 days after the first vaccination. The saliva total Ig antibody responses in mucosal secretion against SARS-CoV-2 RBD were assessed by an in-house-developed assay [15]. Details of the laboratory assays could be found in the Appendix A. In addition, influenza A antibody responses by haemagglutination inhibition (HAI) assay, measured at 28 days after the first vaccination, were evaluated.

### 2.5. Sample Size Calculation

The sample size was determined based on precedence without any formal hypothesis testing. The study enrolled 29 participants (cohort 1:14; cohort 2:15) randomized in a 4:1 ratio such that 11 participants received the low-dose vaccine and 3 participants received the matching placebo in cohort 1, and 12 participants received high-dose and 3 participants received placebo. In addition, this sample size permitted initial estimation of reactogenicity. Given a total of 23 participants who received the test product in two cohorts, the study had an 80% probability of detecting at least 1 drug-related event occurring at a rate of 6.5%.

### 2.6. Statistical Analysis

The number and percentage of participants and the associated exact 95% confidence intervals for adverse events after the receipt of the investigational vaccine or placebo were reported for each study group. Categorical variables and continuous variables were compared using Fisher’s exact test and the Mann-Whitney U test respectively. Medians with interquartile ranges for antibody responses against SARS-CoV-2 were presented. All reported *p*-values were not adjusted for multiple comparisons.

## 3. Results

### 3.1. Participants

Thirty-four healthy Chinese participants were eligible. Five participants decided to withdraw consent before randomization. No replacement of participant was required according to the study protocol. Twenty-nine healthy Chinese participants were vaccinated of which 11 participants received the low-dose vaccine, 12 participants received the high-dose vaccine, and 6 participants received the matching placebo (Table 1; Figure 2). All vaccinated participants completed two doses of DelNS1-nCoV-RBD LAIV and none were discontinued due to an adverse event or COVID-19 infection during the clinical trial period. Twenty participants (69%) were male. The median age was 26 (Interquartile range; IQR 22–37.5) years. The baseline median (IQR) body mass index (BMI) was 22.1 kg/m^2^ (20.8–24.1 kg/m^2^).

### 3.2. Safety Analyses

No serious adverse events or adverse events of special interest were reported within 56 days after the first vaccination in all three groups. No participant discontinued due to adverse events (Table 2). Most reactogenicity reported was mild (Figure 3A,B). Only one participant from the high-dose group complained of grade 3 severity abdominal pain and diarrhoea within 14 days of the first vaccination, which was self-limited (Figure 3A). No subject complained of reactogenicity of grade 3 severity within 14 days of the second vaccination (Figure 3B).

### 3.3. Immunogenicity Analyses

The immunogenicity endpoints included humoral, cellular, and mucosal immune responses post-vaccination. Cellular response induced by the DelNS1-nCoV-RBD LAIV vaccine was evaluated via INF-γ ELISpot. The serum T-cell response measured by IFN-γ ELISpot assay was higher in the high-dose group than the placebo group on day 14 (14 days after the first vaccination) [15 (0–31.3) vs. 0 (0–10) SFU/10^6^ PBMC; *p* = 0.17] and day 42 (14 days after the second vaccination) [12.5 (5–52.5) vs. 5 (0–27.5) SFU/10^6^ PBMC; *p* = 0.18], and also higher in the high-dose group than the low-dose group on day 14 [15 (0–31.3) vs. 0 (0–5) SFU/10^6^ PBMC; *p* = 0.09] and day 42 [12.5 (5–52.5) vs. 0 (0–40) SFU/10^6^ PBMC; *p* = 0.09], despite statistically not reaching significance (Table 3; Figure 4).

The DelNS1-nCoV-RBD LAIV vaccine could elicit mucosal immunity against SARS-CoV-2, as the vaccine was delivered via intranasal rout. To evaluate the mucosal immune response after vaccination, total Ig in saliva was tested by ELISA assay. The saliva total Ig against SARS-CoV-2 RBD of the high-dose vaccine was significantly higher than the control on day 31 (3 days after the second vaccination) [0.24 (0.13–0.63) vs. 0.21 (0.11–0.26); *p* = 0.046] (Table 4; Figure 5). The saliva total Ig of the high-dose and low-dose vaccine were 0.31 (0.09–0.61) and 0.31 (0.07–0.48) respectively, which were twice of that in the placebo group [0.15 (0.06–0.53)] on day 56 after full vaccination.

To determine the humoral response, anti-RBG IgG and a neutralizing antibody in serum were tested by CMIA and live virus MN assays respectively, as the DelNS1-nCoV-RBD LAIV vaccine contained RBD which can elicit an anti-RBD antibody. The serum anti-RBD IgG against SARS-CoV-2 and live virus neutralizing antibody titres were undetectable in all samples (Table 5 and Table 6). The HAI titre measured on day 28 was higher in the high-dose vaccine comparing to the low-dose vaccine and placebo groups [640 (320–1280) vs. 320 (160–640); *p* = 0.29; vs. 240 (160–800); *p* = 0.21], despite statistically not reaching significance (Table 7). There was no difference in the fold-increase.

The error bars represent median with Interquartile ranges (IQR).

## 4. Discussion

This was the first phase 1 clinical trial demonstrating that the DelNS1-nCoV-RBD LAIV for COVID-19 in healthy adults, delivered intranasally is safe and immunogenic. No participants were diagnosed with COVID-19 infection during the clinical trial period, excluding the immunological effects after infection. No serious adverse events or ad-verse events of special interest were reported. The most common adverse events were malaise, myalgia, and sneezing, and were overall mild and self-limiting. This is similar to those reported for the live attenuated intranasal influenza vaccine [16]. Despite no humoral immune response demonstrated, the vaccine elicited sufficient cellular and mucosal response, as demonstrated by a rise and sustainable T-cell response and saliva total Ig against SARS-CoV-2 RBD in the high-dose group, up to 42 and 56 days respectively.

Various studies have demonstrated that current COVID-19 vaccines are unable to prevent nasal SARS-CoV-2 infection and asymptomatic transmission, with a lack of mucosal immunity [3,4,5,17]. More recently, our study in mouse models demonstrated that the intranasal live attenuated influenza-based COVID-19 vaccines (LAIV-CA4-RBD and LAIV-HK68-RBD) based on the same vaccine platform of the current study, and the intramuscular PD1-based receptor-binding domain DNA vaccine (PD1-RBD-DNA) induced satisfactory mucosal and systemic immunity. The induced bronchoalveolar lavage IgA/IgG and lung polyfunctional memory CD8 T cells conferred effective SARS-CoV-2 prevention in both upper and lower respiratory tracts, which cross-neutralized variants of concerns [11]. Therefore, the intranasal vaccine could be used as a booster vaccine for people who have already received the injectable COVID-19 vaccines or to patients who have recovered from a SARS-CoV-2 infection with sufficient humoral and cellular immunity. Other animal studies including the single-dose intranasal chimpanzee adenovirus-vectored vaccine [18], the intranasal helper-dependent adenoviral vector vaccine [19], unadjuvanted intranasal spike vaccine [20], and intranasal administration of RBD nanoparticles induced robust mucosal and systemic immunity against SARS-CoV-2 infection [21]. All these studies demonstrated SARS-CoV-2 specific CD8+ T cell responses, including a high percentage and number of IFN-γ and B cells secreting IgA in the nasal mucosa, trachea, lung, and the spleen.

Both in-vitro and in-vivo studies on the Omicron variants have demonstrated that the infection is characterized by less efficient replication and fusion activity, milder clinical presentation but higher transmission rate [22,23]. Studies have already suggested that neutralizing antibody titre against COVID-19 with the current available vaccines were unable to sustain beyond 6 months [24,25]. It is likely that SARS-CoV-2 infection will surge during the winter period in the northern hemisphere and vaccination against SARS-CoV-2 may be required annually like the seasonal influenza vaccination. Therefore, when compared with other intranasal COVID-19 vaccines based on adenovirus and Newcastle disease virus [9,26,27], the DelNS1-nCoV-RBD LAIV provides an excellent platform to combine both COVID-19 and seasonal influenza vaccination in one single vaccine. This will further improve the vaccination compliance especially in children. The immunological response of the intranasal vaccine in the elderly will need further testing.

In comparison to the aerosolised adenovirus type-5 vector-based COVID-19 vaccine, the DelNS1-nCoV-RBD LAIV is far easier to administer. For the aerosolised vaccination, the vaccine must be administered for 30–60 s using a special nebulization inhalation device [10], during which the vaccine is nebulized and delivered into a disposable mouthpiece. The vaccination procedure must be performed in a single-room facility to prevent the potential risk of environmental contamination. The current vaccine, however, comes in a prefilled syringe like the nasal spray LAIV quadrivalent influenza vaccine and could be administered at home.

There are several limitations of the study. The study size was relatively small and further study on the effect of the higher dose is needed. We could only perform a total Ig mucosal assay in lieu of IgA mucosal assay as stipulated in the study protocol due to interference. A future assay will be developed to measure solely the IgA level to better assess the mucosal immunity. The T-cell responses on day 7 and 28 in a few cases in the placebo group could be non-specific effects which are markedly lower than the high dose group. The study was unable to demonstrate a humoral response which could be overcome by a future booster dose on people who have already received COVID-19 vaccination or have recovered from an infection. Assessment of the long-term immunity and protection against SARS-CoV-2 infection conferred by the intranasal vaccine is also needed.

## 5. Conclusions

The high-dose intranasal DelNS1-nCoV-RBD LAIV is safe with moderate mucosal immunogenicity. As most subjects have already received COVID-19 vaccination, a phase-2 booster trial with a two-dose regimen of the high-dose intranasal DelNS1-nCoV-RBD LAIV is warranted.

## Figures and Tables

**Figure 1 vaccines-11-00723-f001:**
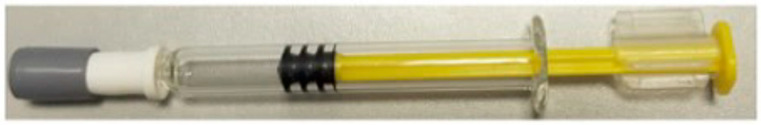
A sample of the intranasal DelNS1-nCoV-RBD LAIV.

**Figure 2 vaccines-11-00723-f002:**
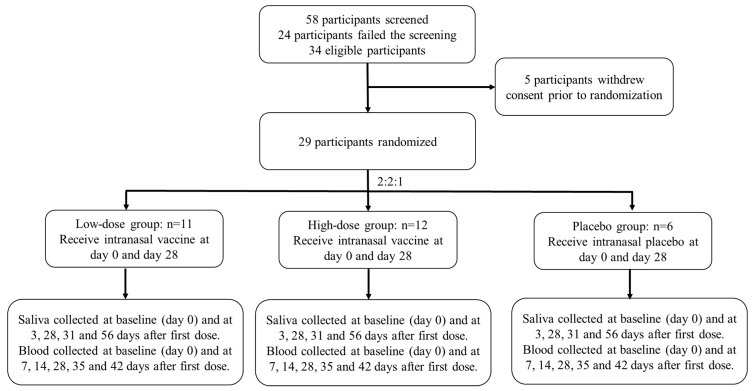
Overview of participants’ allocation. High-dose: DelNS1-nCoV-RBD LAIV (1 × 10^7.7^ EID_50_/dose) in 0.2 mL. Low-dose: DelNS1-nCoV-RBD LAIV (1 × 10^7^ EID_50_/dose) in 0.2 mL. Placebo: inert excipients/dose in 0.2 mL.

**Figure 3 vaccines-11-00723-f003:**
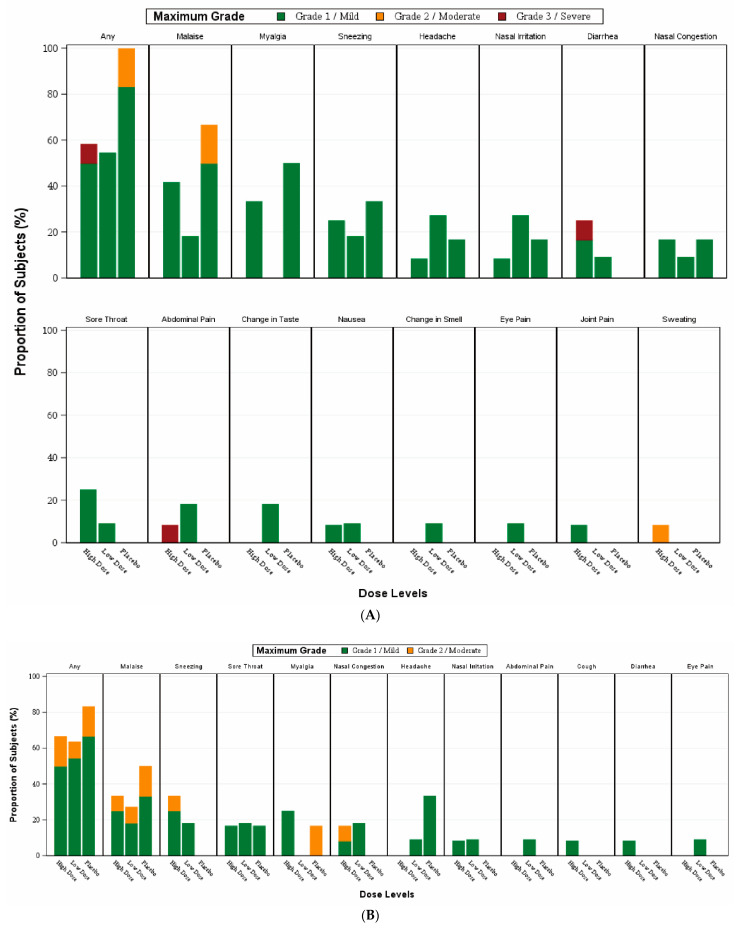
Reactogenicity events reported for the 14-day period after each vaccination with DelNS1-nCoV-RBD LAIV. Subjects received two doses of test vaccine or placebo on day 1 and day 29, and were then required to record any AE in the diary for a 14-day period after each vaccination. (**A**) AE occurred in subjects after the first vaccination. (**B**) AE occurred in subjects after the second dose. Fisher’s exact test was used to analyze the AEs in the three groups.

**Figure 4 vaccines-11-00723-f004:**
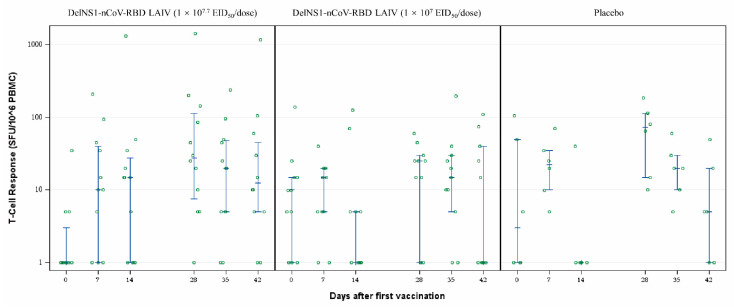
T-cell responses by ELISpot over time. Blood samples were taken from the subjects on baseline (day 0) and at 7, 14, 28, 35, and 42 days after the first dose of vaccine. Once separated from the blood, PBMC were seeded into anti-human IFN-γ antibody coated plate followed by the incubation with SARS-CoV-2 RBD peptide pool overnight. The substrate was added into the plate after cells incubated with Streptavidin-Alkaline Phosphatase (ALP). The spots in the plate were counted under an immunospot reader. One-way ANOVA was used to compare the T-cell response in the three groups. The error bars represent median with interquartile ranges (IQR).

**Figure 5 vaccines-11-00723-f005:**
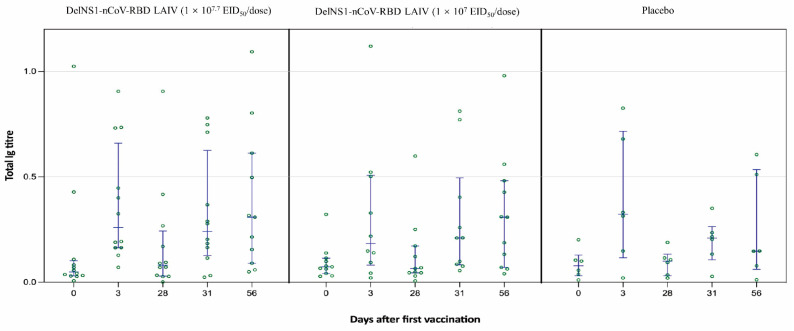
Total saliva Ig titre over time. Saliva samples were collected at baseline (day 0) and at 3, 28, 31, and 56 days after the first vaccination. After being treated with 1% Triton-100 for 30 min, saliva was diluted 2 folds with PBS. Biotinylated recombinant SARS-CoV-2 RBD was added into the 96-well plate coated with avidin for a 30 min-incubation. Then, treated saliva samples were added into the wells. After 1 h, the plate was incubated with horseradish peroxidase (HRP) conjugated goat anti-human IgG, IgM, and IgA antibody. Finally, the optical density (OD) was read at 450 and 620 nm. One-way ANOVA was used to compare the mucosal Ig response in the three groups.

**Table 1 vaccines-11-00723-t001:** Demographic and baseline characteristics.

	High-Dose (n = 12)	Low-Dose (n = 11)	Placebo (n = 6)
Age (median; IQR)	25 (21, 35)	25 (22, 34)	38 (26, 45)
Male sex (%)	8 (66.7)	8 (72.7)	4 (66.7)
BMI (kg/m^2^) (median; IQR)	21.9 (20.1, 23.7)	22.3 (21.4, 24.3)	22.3 (19.5, 24.1)

High-dose: DelNS1-nCoV-RBD LAIV (1 × 10^7.7^ EID_50_/dose) in 0.2 mL. Low-dose: DelNS1-nCoV-RBD LAIV (1 × 10^7^ EID_50_/dose) in 0.2 mL. Placebo: inert excipients/ dose in 0.2 mL. IQR: interquartile range; BMI: body mass index.

**Table 2 vaccines-11-00723-t002:** Summary of adverse events within 56 days after the first vaccination.

	High-Dose (n = 12)	Low-Dose (n = 11)	Placebo (n = 6)	*p*-Value +
Subjects with reactogenicities	9 (75%) {54}	8 (72.7%) {44}	6 (100%) {19}	0.60
95% CI	(42.81– 94.51%)	(39.03–93.98%)	(54.07–100%)	
Subjects with unsolicited adverse events	9 (75%) {15}	6 (54.5%) {17}	4 (66.7%) {7}	0.62
95% CI	(42.81–94.51%)	(23.38–83.25%)	(22.28–96.67%)	
Subjects with adverse events of special interest (AESIs)	0	0	0	-
95% CI	-	-	-	-
Subjects with serious adverse events (SAEs)	0	0	0	-
95% CI	-	-	-	-
Subjects discontinued due to adverse events	0	0	0	-
95% CI	-	-	-	-

High-dose: DelNS1-nCoV-RBD LAIV (1 × 10^7.7^ EID_50_/dose) in 0.2 mL. Low-dose: DelNS1-nCoV-RBD LAIV (1 × 10^7^ EID_50_/dose) in 0.2 mL. Placebo: inert excipients/ dose in 0.2 mL. + Fisher’s exact test. CI: confidence interval; ( ) = percentage of subjects with adverse events; { } = number of adverse events.

**Table 3 vaccines-11-00723-t003:** Summary of T-Cell ELISpot response.

	High-Dose (n = 12)	Low-Dose (n = 11)	Placebo (n = 6)	*p*-Value ^1+^	*p*-Value ^2+^	*p*-Value ^3+^	*p*-Value ^4^
Day 0 (pre-dose)	0 (0–3.8)	10 (0–15)	2.5 (0–63.8)	0.09	0.46	0.10	0.29
Day 7	10 (0–42.5)	15 (5–20)	22.5 (8.8–43.8)	0.23	0.17	0.03	0.45
Day 14	15 (0–31.3)	0 (0–5)	0 (0–10)	0.17	0.16	0.09	0.52
Day 28 (pre-dose)	27.5 (6.3–130)	25 (0–30)	72.5 (13.8–132.5)	0.55	0.08	0.29	0.43
Day 35	20 (5–48.8)	15 (5–30)	20 (8.75–37.5)	0.19	0.38	0.55	0.80
Day 42	12.5 (5–52.5)	0 (0–40)	5 (0–27.5)	0.18	0.19	0.09	0.49

High-dose: DelNS1-nCoV-RBD LAIV (1 × 10^7.7^ EID_50_/dose) in 0.2 mL. Low-dose: DelNS1-nCoV-RBD LAIV (1 × 10^7^ EID_50_/dose) in 0.2 mL. Placebo: inert excipients/ dose in 0.2 mL. *p*-value ^1^: High-dose vs. Placebo. *p*-value ^2^: Low-dose vs. Placebo. *p*-value ^3^: High-dose vs. Low-dose. + Mann-Whitney U test. *p*-value ^4^: one-way ANOVA, comparison of three groups. data were median SFU/10^6^ PBMC (IQR).

**Table 4 vaccines-11-00723-t004:** Summary of saliva total Ig response.

	High-Dose (n = 12)	Low-Dose (n = 11)	Placebo (n = 6)	*p*-Value ^1+^	*p*-Value ^2+^	*p*-Value ^3+^	*p*-Value ^4^
Day 0 (pre-dose)	0.05 (0.03–0.10)	0.07 (0.04–0.11)	0.08 (0.03–0.13)	0.15	0.93	0.06	0.65
Day 3	0.26 (0.16–0.66)	0.18 (0.08–0.51)	0.32 (0.12–0.72)	0.83	0.99	0.84	0.87
Day 28 (pre-dose)	0.08 (0.03–0.24)	0.07 (0.04–0.17)	0.1 (0.03–0.13)	0.11	0.20	0.36	0.66
Day 31	0.24 (0.13–0.63)	0.21 (0.08–0.50)	0.21 (0.11–0.26)	0.046	0.06	0.99	0.59
Day 56	0.31 (0.09–0.61)	0.31 (0.07–0.48)	0.15 (0.06–0.53)	0.45	0.96	0.45	0.69

High-dose: DelNS1-nCoV-RBD LAIV (1 × 10^7.7^ EID_50_/dose) in 0.2 mL. Low-dose: DelNS1-nCoV-RBD LAIV (1 × 10^7^ EID_50_/dose) in 0.2 mL. Placebo: inert excipients/ dose in 0.2 mL. *p*-value ^1^: High-dose vs. Placebo. *p*-value ^2^: Low-dose vs. Placebo. *p*-value ^3^: High-dose vs. Low-dose. + Mann-Whitney U test. *p*-value ^4^: one-way ANOVA, comparison of three groups. Data were median (IQR).

**Table 5 vaccines-11-00723-t005:** Summary of anti-RBD IgG titre in serum.

	High-Dose (n = 12)	Low-Dose (n = 11)	Placebo (n = 6)
Day 0 (pre-dose)	0 (0–0.6)	8.5 (0.5–18.7)	1.5 (0–5.1)
Day 7	0 (0–1.1)	4.0 (0–7.2)	0.4 (0–2.9)
Day 14	0 (0–0.6)	2.3 (0–11.4)	0.65 (0–5.9)
Day 28 (pre-dose)	0 (0–0.8)	2.0 (0–4.4)	0 (0–1.8)
Day 35	0.3 (0–1.3)	3.9 (0–8.4)	2.0 (0–3.6)
Day 42	0 (0–0.8)	3.7 (0.2–7.3)	2.8 (0–5.0)
Day 56	0 (0–0.8)	2.5 (0.8–4.4)	4.8 (0–6.8)

Date are median (IQR); Unit: AU/mL. High-dose: DelNS1-nCoV-RBD LAIV (1 × 10^7.7^ EID_50_/dose) in 0.2 mL. Low-dose: DelNS1-nCoV-RBD LAIV (1 × 10^7^ EID_50_/dose) in 0.2 mL. Placebo: inert excipients/dose in 0.2 mL. Anti-RBD IgG titre ≥ 50 AU/mL was considered as positive.

**Table 6 vaccines-11-00723-t006:** Summary of live virus microneutralizing titre (MN titre).

	High-Dose (n = 12)	Low-Dose (n = 11)	Placebo (n = 6)
Day 0 (pre-dose)	5 (5–5)	5 (5–5)	5 (5–5)
Day 7	5 (5–5)	5 (5–5)	5 (5–5)
Day 14	5 (5–5)	5 (5–5)	5 (5–5)
Day 28 (pre-dose)	5 (5–5)	5 (5–5)	5 (5–5)
Day 35	5 (5–5)	5 (5–5)	5 (5–5)
Day 42	5 (5–5)	5 (5–5)	5 (5–5)
Day 56	5 (5–5)	5 (5–5)	5 (5–5)

Date are geometric mean titre (95% confidence interval). High-dose: DelNS1-nCoV-RBD LAIV (1 × 10^7.7^ EID_50_/dose) in 0.2 mL. Low-dose: DelNS1-nCoV-RBD LAIV (1 × 10^7^ EID_50_/dose) in 0.2 mL. Placebo: inert excipients/ dose in 0.2 mL. MN titre ≥ 10 was considered as positive.

**Table 7 vaccines-11-00723-t007:** Summary of influenza A H1N1 haemagglutination inhibition assay.

	High-Dose (n = 12)	Low-Dose (n = 11)	Placebo (n = 6)	*p*-Value ^1+^	*p*-Value ^2+^	*p*-Value ^3+^	*p*-Value ^4^
Day 0 (pre-dose)	320 (160–640)	320 (80–640)	160 (80–280)	0.13	0.93	0.06	0.40
Day 28 (pre-dose)	640 (320–1280)	320 (160–640)	240 (160–800)	0.21	0.29	0.36	0.34
Fold-increase	2 (1–2)	1 (1–2)	1.5 (1–5.5)	0.75	0.10	0.99	0.24

High-dose: DelNS1-nCoV-RBD LAIV (1 × 10^7.7^ EID_50_/dose) in 0.2 mL. Low-dose: DelNS1-nCoV-RBD LAIV (1 × 10^7^ EID_50_/dose) in 0.2 mL. Placebo: inert excipients/ dose in 0.2 mL. *p*-value ^1^: High-dose vs. Placebo. *p*-value ^2^: Low-dose vs. Placebo. *p*-value ^3^: High-dose vs. Low-dose. + Mann-Whitney U test. *p*-value ^4^: one-way ANOVA, comparison of three groups. HAI: haemagglutination inhibition assay; IQR: interquartile range. Data were median HAI titre (IQR).

## Data Availability

The data used to support the findings of this study are included within the article.

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
