# Peer review of "A Phase 1, Randomized, Double-Blinded, Placebo-Controlled and Dose-Escalation Study to Evaluate the Safety and Immunogenicity of the Intranasal DelNS1-nCoV-RBD LAIV for COVID-19 in Healthy Adults"

_vaccines, 2023, doi:10.3390/vaccines11040723_

Round 1

Reviewer 1 Report

This paper by Ruiqi Zhang et al. describes a phase 1 dose-escalation study designed to test the safety and the efficacy of a COVID-19 intranasal vaccine. A cohort of 29 participants were randomized in a low dose group, a high dose group and a placebo group in a 2:2;1 ratio. No serious adverse events were reported by any of the study participants within the first two months from the first vaccination. Hence, the vaccine appears to be safe at the tested dosage. The efficacy of the vaccine was assessed by measuring the level of mucosal Ig. Results suggest some degree of immunogenicity when a high dose of the vaccine is administered.

Overall, this phase 1 study appears to provide some evidence that this intranasal vaccine may find potential use as a booster for individuals who have already received a COVID-19 vaccination, even though further studies on larger cohorts and with different doses are clearly warranted.

Overall, I think that this manuscript can be accepted in Vaccines. However, some minor changes to the text are needed before the manuscript can be accepted for publication:

1)     Line31: “The low and high-dose vaccines were composed of…..”

2)     Line 32: “The placebo was composed of…..”

3)     Line46: “….after receiving….”

4)     Line 60: “promising preventive measures to contain….”

5)     Line 86: “….influenza virus or other viruses”

6)     Line 88: “…elicits an immune response….

7)     Line 88: ….., which is an important antigen….”

8)     Line 255: “….in the three groups.”

9)     Line 277: “…the T cell response in the three groups.”

10) Line 277: “…the T cell response in the three groups.”

11) Line 310: “….Ig response in the three groups.”

12) Line 371: “It is likely that SARS-CoV-2 infection….”

13) Line 372: please remove “annual”, as it is repeated in the following line. Perhaps, you can replace “annual” with “that”: “…in the northern hemisphere and that vaccination against SARS-CoV-2 may be required annually like the seasonal influenza vaccination”.

Fig. 3: In both panels (A and B), I would increase the separation between the two blocks of diagrams a little, to improve clarity with respect to the title referring to each type of events.

Author Response

We are grateful to the helpful comments given by the Referee. We have carefully revised the manuscript accordingly. A point-to-point response to Reviewer‘s comments is given as follows.

This paper by Ruiqi Zhang et al. describes a phase 1 dose-escalation study designed to test the safety and the efficacy of a COVID-19 intranasal vaccine. A cohort of 29 participants were randomized in a low dose group, a high dose group and a placebo group in a 2:2;1 ratio. No serious adverse events were reported by any of the study participants within the first two months from the first vaccination. Hence, the vaccine appears to be safe at the tested dosage. The efficacy of the vaccine was assessed by measuring the level of mucosal Ig. Results suggest some degree of immunogenicity when a high dose of the vaccine is administered.

Overall, this phase 1 study appears to provide some evidence that this intranasal vaccine may find potential use as a booster for individuals who have already received a COVID-19 vaccination, even though further studies on larger cohorts and with different doses are clearly warranted.

Overall, I think that this manuscript can be accepted in Vaccines. However, some minor changes to the text are needed before the manuscript can be accepted for publication:

1)     Line 31: “The low and high-dose vaccines were composed of…..”

Response:We thank the Reviewer for the suggestion, and we have amended the sentence accordingly.

Line 31-32: The low and high-dose vaccine were composed of 1x 10^7 EID50/ dose and 1x 10^7.7 EID50/ dose in 0.2mL respectively.

2)     Line 32: “The placebo was composed of…..”

Response:We thank the Reviewer for the suggestion, and we have revised the sentence accordingly.

Line 33: The placebo vaccine was composed of inert excipients/dose in 0.2mL.

3)     Line46: “….after receiving….”

Response:We thank the Reviewer for the comment, and we have amended the sentence accordingly.

Line 45-46:  The high-dose group showed slightly higher level of mucosal Ig than control group after receiving two doses of vaccine (day 35, 0.24 vs. 0.21, p=0.046; day 56 0.31 vs. 0.15, p=0.45).

4)     Line 60: “promising preventive measures to contain….”

Response:We thank the Reviewer for the suggestion, and we have amended the sentence accordingly.

Line 60:  With the lack of specific antiviral treatment, vaccines remain the most promising preventive measures to contain this pandemic. 

5)     Line 86: “….influenza virus or other viruses”

Response:We thank the Reviewer for the suggestion, and we have revised the sentence accordingly.

Line 85-86: Compared with traditional LAIV, DelNS1-LAIV can work as a vector to express antigens from influenza virus or other viruses [10].

6)     Line 88: “…elicits an immune response….

Response:We thank the Reviewer for the suggestion, and we have amended the sentence accordingly.

Line 88: The COVID-19 vaccine elicits an immune response against SARS-CoV-2.

7)     Line 88: ….., which is an important antigen….”

Response:We thank the Reviewer for the comment, and we have added “an” to the sentence.

Line 88-89: the expression of RBD which is an important antigen of SARS-CoV-2 and target of neutralizing antibody.

8)     Line 255: “….in the three groups.”

Response:We thank the Reviewer for the suggestion, and we have added “the” to the sentence.

Line 262: Fisher’s exact test was used to analyze the AEs in the three groups.

9)     Line 277: “…the T cell response in the three groups.”

Response:We thank the Reviewer for the suggestion, and we have added “the” to the sentence.

Line 286: One-way ANOVA was used to compare the T cell response in the three groups.

10) Line 277: “…the T cell response in the three groups.”

Response:We thank the Reviewer for the suggestion, and we have added “the” to the sentence.

Line 286: One-way ANOVA was used to compare the T cell response in the three groups.

11) Line 310: “….Ig response in the three groups.”

Response:We thank the Reviewer for the suggestion, and we have added “the” to the sentence.

Line 328: One-way ANOVA was used to compare the mucosal Ig response in the three groups.

12) Line 371: “It is likely that SARS-CoV-2 infection….”

Response:We thank the Reviewer for the suggestion, and we have amended the sentence accordingly.

Line 407-409: It is likely that SARS-CoV-2 infection will surge during the winter period in the northern hemisphere and vaccination against SARS-CoV-2 may be required annually like the seasonal influenza vaccination.

13) Line 372: please remove “annual”, as it is repeated in the following line. Perhaps, you can replace “annual” with “that”: “…in the northern hemisphere and that vaccination against SARS-CoV-2 may be required annually like the seasonal influenza vaccination”.

Response:We thank the Reviewer for the suggestion, and we have deleted “annual” from the sentence.

Line 407-409: It is likely that SARS-CoV-2 infection will surge during the winter period in the northern hemisphere and vaccination against SARS-CoV-2 may be required annually like the seasonal influenza vaccination.

Fig. 3: In both panels (A and B), I would increase the separation between the two blocks of diagrams a little, to improve clarity with respect to the title referring to each type of events.

Response: We thank the Reviewer for the suggestion, and we have increased the distance between the two blocks in Figure 3.

Reviewer 2 Report

The manuscript “A Phase 1, Randomized, Double-blinded, Placebo-controlled and Dose-escalation Study to Evaluate the Safety and Immunogenicity of the Intranasal DelNS1-nCoV-RBD LAIV for COVID-19 in Healthy Adults” assessed safety and immunogenicity of the COVID-19 vaccine in healthy adults.

The manuscript is clear and well organized and the manuscript might be accepted for publications after the following points are duly addressed:

Major points:

1.    To study immunogenicity, the authors used IFN ELISpot assay. From reference 14, “PMA/ionomycin treatment was used as the positive control and anti-CD28/anti-CD49d mAbs treatment was used as the as the negative control. The ELISPOT assay was performed using the human IFN-γ ELISPOT Kit (Mabtech) according the manufacturer’s instructions. Spots were counted using an immunospot reader and image analyzer (Cellular Technology Limited). Results were considered positive when the number of spot-forming cells (SFC)/106 PBMCs was 2-fold above that of the negative controls.” Do the authors include similar positive/negative control in their experiment? If so, please state in the method section.

2.    In Figure 4 and 5. Could the high variance in the measurements be explained? Specifically, please include explanation in the section 3.3, the high variance for placebo group within same day, more importantly, the high variance among days.

3.    Similar to the point above, for Line 261, “the serum T-cell response measured by IFN ELISpot assay was higher in the high-dose group than the placebo group on day 14 and day 42”. It is true for Day 14, and Day 42. But because of the high variance, high-dose group is lower than placebo group for the other days. Could the author also provide explanation for this?

4.    Similar to the point above, measurement of Total saliva Ig titre has high variance. The highest median response comes from Day 3, placebo group, not from the high dose group. Could the author also provide explanation for this?

Minor points:

1.    Figure 5, y axis, the unit seems to be missing. Is it AU/mL?

Author Response

We are grateful to the helpful comments given by Referee. We have carefully revised the manuscript accordingly. A point-to-point response to Reviewer’s comments is given as follows.

Reviewer :

The manuscript “A Phase 1, Randomized, Double-blinded, Placebo-controlled and Dose-escalation Study to Evaluate the Safety and Immunogenicity of the Intranasal DelNS1-nCoV-RBD LAIV for COVID-19 in Healthy Adults” assessed safety and immunogenicity of the COVID-19 vaccine in healthy adults.

The manuscript is clear and well organized and the manuscript might be accepted for publications after the following points are duly addressed:

Major points:

  1. To study immunogenicity, the authors used IFN ELISpot assay. From reference 14, “PMA/ionomycin treatment was used as the positive control and anti-CD28/anti-CD49d mAbs treatment was used as the as the negative control. The ELISPOT assay was performed using the human IFN-γ ELISPOT Kit (Mabtech) according the manufacturer’s instructions. Spots were counted using an immunospot reader and image analyzer (Cellular Technology Limited). Results were considered positive when the number of spot-forming cells (SFC)/106 PBMCs was 2-fold above that of the negative controls.” Do the authors include similar positive/negative control in their experiment? If so, please state in the method section.

Response: We thank the reviewer for the comments. we used the same ELISpot protocol as reference 14, and we have added the information about positive and negative control to the Method.

Line 178-180: The PMA/ionomycin was used as the positive control, while anti-CD28/anti-CD49d mAbs treatment as the negative control.

  1. In Figure 4 and 5. Could the high variance in the measurements be explained? Specifically, please include explanation in the section 3.3, the high variance for placebo group within same day, more importantly, the high variance among days.

Response: We thank the reviewer for the comments. The error bars in Figure 4 and 5 represent median with interquartile ranges (IQR), and we have added the information to the figure legends. The large IQR could be caused by the small sample size. Furthermore, non-specific reaction in experiment could be another reason.

Line 282-289: Figure 4. T cell responses by ELISpot over time. Blood samples were taken from the subjects on baseline (day 0) and at 7, 14, 28, 35, and 42 days after the first dose of vaccine. After separated from the blood, PBMC were seeded into anti-human IFN-γ antibody coated plate followed by the incubation with SARS-CoV-2 RBD peptide pool for over-night. The substrate was added into the plate after cells incubated with Streptavidin- Alkaline Phosphatase (ALP). The spots in the plate were counted under an immunospot reader. One-way ANOVA was used to compare the T cell response in the three group. The error bars represent median with interquartile ranges (IQR).

Line 323-332: Figure 5. Total saliva Ig titre over time. Saliva samples were collected at baseline (day 0) and at 3, 28, 31 and 56 days after the first vaccination. After treated with 1% Triton-100 for 30 min, saliva was diluted 2 folds with PBS.  Biotinylated recombinant SARS CoV-2 RBD was added into the 96-well plate coated with avidin for a 30 min-incubation. Then, treated saliva samples were added into the wells. After 1 hour, the plate was incubated with horseradish peroxidase (HRP) conjugated goat anti-human IgG, IgM and IgA antibody. Finally, the optical density (OD) was read at 450 and 620 nm. One-way ANOVA was used to compare the mucosal Ig response in the three group. The error bars represent median with interquartile ranges (IQR).

  1. Similar to the point above, for Line 261, “the serum T-cell response measured by IFN ELISpot assay was higher in the high-dose group than the placebo group on day 14 and day 42”. It is true for Day 14, and Day 42. But because of the high variance, high-dose group is lower than placebo group for the other days. Could the author also provide explanation for this?

Response: We thank the reviewer for the comment. The large IQR could be caused by the small sample size and non-specific reactions in experiment. Despite higher T cell response observed in placebo group in comparison with high-dose group at day 0, 7, 28 and 35, there was no significant difference. Similarly, the T cell response at day 14 and 42 was higher in high-dose group than that in placebo group with no significant difference.

  1. Similar to the point above, measurement of Total saliva Ig titre has high variance. The highest median response comes from Day 3, placebo group, not from the high dose group. Could the author also provide explanation for this?

Response: We thank the reviewer for the question. A possible explanation was the quality of saliva samples as these samples were self-collected by individual participant. Moreover, the non-specific binding in the experiment and small size of placebo group could also result in the highest median of saliva Ig from placebo group. Although the day 3 saliva Ig in placebo group was higher than vaccine groups, no significant difference was observed.  

Minor points:

  1. Figure 5, y axis, the unit seems to be missing. Is it AU/mL?

Response: We thank the reviewer for the comment. As the data about saliva Ig are OD value, there is no unit.

Reviewer 3 Report

This manuscript reports on the result of a phase I clinical trial for a novel COVID-19 vaccine which is based on the adaptation of a LAIV flu virus.

This is an interesting approach, and if successful could provide an extremely useful vaccine for the continued prevention of this important disease.

The study is well designed and carried out, though somewhat limited in scope at only 29 participants. However, these limitations are acknowledged by the authors and judged to be sufficient to identify adverse effects occurring at ~6% with an 80% accuracy.

In the study no significant adverse effects were detected. However, the levels of immunogenicity generated by the vaccine appear to be low, often of no statistical significance overall.

In general the paper is well written and the data clearly presented. The conclusions drawn are supported by the data and importantly the authors freely acknowledge the limitations of the study.

Therefore, my opinion is that this manuscript is suitable for publication in vaccines.

I agree with the authors that this vaccine should be continued to a phase II trial, and await the result with interest.

Author Response

We are grateful to the helpful comments given by Referee. We have carefully revised the manuscript accordingly. A point-to-point response to Reviewer’s comments is given as follows.

Reviewer:

This manuscript reports on the result of a phase I clinical trial for a novel COVID-19 vaccine which is based on the adaptation of a LAIV flu virus.

This is an interesting approach, and if successful could provide an extremely useful vaccine for the continued prevention of this important disease.

The study is well designed and carried out, though somewhat limited in scope at only 29 participants. However, these limitations are acknowledged by the authors and judged to be sufficient to identify adverse effects occurring at ~6% with an 80% accuracy.

In the study no significant adverse effects were detected. However, the levels of immunogenicity generated by the vaccine appear to be low, often of no statistical significance overall.

In general the paper is well written and the data clearly presented. The conclusions drawn are supported by the data and importantly the authors freely acknowledge the limitations of the study.

Therefore, my opinion is that this manuscript is suitable for publication in vaccines.

I agree with the authors that this vaccine should be continued to a phase II trial, and await the result with interest.

Response: We thank the Reviewer for the comments and decision.  

Reviewer 4 Report

vaccines-2228525 

The manuscript presents a phase 1 clinical trial to study the effect of an intranasal COVID-19 vaccine. Although a highly significant topic, the work and its presentation have several flaws in its content and description. Specific comments are as follows:

1.     How the 2 vaccine doses were chosen? 

2.     Why the detection of Abs was not tested in serum? That must be shown.

3.     A major weakness is that this vaccine induced a modest mucosal immunity

4.     An overview of the vaccine treatment, samples collected should be included.

5.     The description of the data are not clear. The rationale for assessing each parameter is not mentioned.

6.     Some of the values between the figures and tables do not match.

7.     Figure 3 shows the same graph is different panels.

8.     Multiple intranasal vaccines against SARS-CoV-2 have been designed. However, the discussion is limited to include 2 related references.

Minor comments:

9.     The abstract is not clear to indicate from the beginning that this is a vaccine against SARS-CoV-2. It rather gives emphasis to the vector.

10.  Presentation of the tables needs improvement. The labels are confusing. 

11.  Supplementary files were not compatible to open the file.

Author Response

We are grateful to the helpful comments given by Referee. We have carefully revised the manuscript accordingly. A point-to-point response to Reviewer’s comments is given as follows.

Reviewer:

vaccines-2228525 

The manuscript presents a phase 1 clinical trial to study the effect of an intranasal COVID-19 vaccine. Although a highly significant topic, the work and its presentation have several flaws in its content and description. Specific comments are as follows:

  1. How the 2 vaccine doses were chosen? 

Response: We thank Reviewer for the question. As DelNS1-nCoV-RBD LAIV vaccine is based on live attenuated influenza virus, we choose the dose of the intranasal vaccine by taking the reference to the dose of live attenuated influenza virus vaccine. As shown in previous clinical trials about live attenuated influenza virus vaccine, the dose is no less than 10^7 EID50 [1,2]. In the study, the low-dose vaccine is composed of 10^7 EID50 per dose. For the high-dose vaccine (10^7.7 EID50), since there was no available information about the safety of the intranasal DelNS1-nCoV-RBD LAIV vaccine in human before, the dose of the vaccine was increased moderately to make sure that it was safe for the subjects in the phase I trial. In other phase I studies, the difference in dose level between high-dose and low dose is also moderate and about < 5 folds [3,4].

  1. Why the detection of Abs was not tested in serum? That must be shown.

Response: We thank Reviewer for the question. The antibody in serum had been tested with live virus microneutralization assay and chemiluminescent microparticle immunoassay, and results were presented in table 5 and 6.

  1. A major weakness is that this vaccine induced a modest mucosal immunity.

Response: We thank Reviewer for the comment. In the study, the intranasal COVID-19 vaccine just elicited moderate mucosal immunity. The possible explanation could be that the dose of the vaccine is not high enough to induce a robust mucosal immune response. Furthermore, the pre-existing anti-influenza immunity in subjects impacts the efficacy of the intranasal vaccine. We have revised the manuscript accordingly.

Line 49-50: The high-dose intranasal DelNS1-nCoV-RBD LAIV is safe with significant moderate mucosal immunogenicity.  

Line 438: The high-dose intranasal DelNS1-nCoV-RBD LAIV is safe with significant moderate mucosal immunogenicity.

  1. An overview of the vaccine treatment, samples collected should be included.

Response: We thank Reviewer for the comment, and we have revised the figure 2 accordingly.

  1. The description of the data are not clear. The rationale for assessing each parameter is not mentioned.

Response: We thank Reviewer for the comment and suggestion, and we have added more information about each parameter to Results.

Line 270-272: Cellular response induced by the DelNS1-nCoV-RBD LAIV vaccine was evaluated via INF-γELISpot.

Line 303-305: The DelNS1-nCoV-RBD LAIV vaccine could elicit mucosal immunity against SARS-CoV-2, as the vaccine was delivered via intranasal rout. To evaluate the mucosal immune response after vaccination, total Ig in saliva was tested by ELISA assay.

Line 311-313: To determine the humoral response, anti-RBG IgG and neutralizing antibody in serum were tested by CMIA and live virus MN assays respectively, as the DelNS1-nCoV-RBD LAIV vaccine contained RBD which can elicit anti-RBD antibody.

  1. Some of the values between the figures and tables do not match.

Response: We thank Reviewer for the comment. In figure 4, to display T cell response results over a very wide range of values, we used a logarithmic y axis, which cannot contain zero, in the graph.  Furthermore, as the value of T cell response titre is integer, y axis started at 1. To display samples with T cell response titre=0, the value of T cell response for these samples were considered as 1. In the study, T cell response titre =1 was also considered as negative.

  1. Figure 3 shows the same graph is different panels.

Response: We thank Reviewer for the comments, and we have corrected it.

  1. Multiple intranasal vaccines against SARS-CoV-2 have been designed. However, the discussion is limited to include 2 related references.

Response: We thank Reviewer for the suggestion, and we have cited four more reference about intranasal vaccine in Discussion.

Line 398-402: Other animal studies including the single-dose intranasal chimpanzee adenovi-rus-vectored vaccine [18], the intranasal helper-dependent adenoviral vector vaccine [19], unadjuvanted intranasal spike vaccine [20], and intranasal administration of RBD nanoparticles induced robust mucosal and systemic immunity against SARS-CoV-2 infection [21].

Line 412-415: Therefore, when compared with other intranasal COVID-19 vaccine based on adeno-virus and newcastle disease virus [9,26,27], the DelNS1-nCoV-RBD LAIV provides an excellent platform to combine both COVID-19 and seasonal influenza vaccination in one single vaccine.

Minor comments:

  1. The abstract is not clear to indicate from the beginning that this is a vaccine against SARS-CoV-2. It rather gives emphasis to the vector.

Response: We thank Reviewer for the comments, and we have revised the first sentence to indicate that the intranasal vaccine designed for COVID-19 in Abstract.

 Line 26-27: An intranasal COVID-19 vaccine, DelNS1-based RBD vaccines composed of H1N1 subtype (DelNS1-nCoV-RBD LAIV) was developed to evaluate the safety and immunogenicity in healthy adults.

  1. Presentation of the tables needs improvement. The labels are confusing. 

Response: We thank Reviewer for the comments, and we have amended table2, 3 and 4.

  1. Supplementary files were not compatible to open the file.

Response: We thank Reviewer for the comments, and we have uploaded the supplementary method and protocol in PDF separately.

Reference:

  1. White WG, Freestone DS, BowkerCH, et al. A clinical trial of WRL 105 strain live attenuated influenza vaccine comparing four methods of intranasal vaccination. Dev Biol Stand. 1976;33:202-6.
  2. Nachbagauer R, Feser J, Naficy A, et al. A chimeric hemagglutinin-based universal influenza virus vaccine approach induces broad and long-lasting immunity in a randomized, placebo-controlled phase I trial. Nat Med. 2021 Jan;27(1):106-114.
  3. Wu S, Huang J, Zhang Z, et al. Safety, tolerability, and immunogenicity of an aerosolised adenovirus type-5 vector-based COVID-19 vaccine (Ad5-nCoV) in adults: preliminary report of an open-label and randomised phase 1 clinical trial. Lancet Infect Dis. 2021; 21(12): 1654–1664.
  4. Mulligan MJ, Lyke KE, Kitchin N, et al. Phase I/II study of COVID-19 RNA vaccine BNT162b1 in adults. Nature. 2021;590(7844):E26.

Round 2

Reviewer 4 Report

The manuscript will benefit of an additional round of editing.